# Biological and Cellular Properties of Advanced Platelet-Rich Fibrin (A-PRF) Compared to Other Platelet Concentrates: Systematic Review and Meta-Analysis

**DOI:** 10.3390/ijms25010482

**Published:** 2023-12-29

**Authors:** Vinicius Balan Santos Pereira, Carlos Augusto Pereira Lago, Renata de Albuquerque Cavalcanti Almeida, Davi da Silva Barbirato, Belmiro Cavalcanti do Egito Vasconcelos

**Affiliations:** Oral and Maxillofacial Surgery, Department of Oral and Maxillofacial Surgery and Traumatology, University of Pernambuco, Recife 50100-130, Brazil; viniciusbalan99@gmail.com (V.B.S.P.); carlos.lago@upe.br (C.A.P.L.); renata.almeida@upe.br (R.d.A.C.A.); davibarbirato@gmail.com (D.d.S.B.)

**Keywords:** blood platelets, fibrin, platelet-rich plasma, platelet-rich fibrin, platelet concentrates

## Abstract

Platelet concentrates are used for cell induction and stimulation in tissue repair processes. The aim of the present systematic review and meta-analysis was to compare the biological and cellular properties of advanced platelet-rich fibrin (A-PRF) to those of other platelet concentrates. Searches were conducted on the PubMed/Medline, Scopus, Web of Science, Embase and LILACS databases using a search strategy oriented by the guiding question. A total of 589 records were retrieved. Seven articles of in vitro experimental studies were selected for qualitative data analysis and four were selected for meta-analysis. The release of growth factors, distribution of cells in the fibrin membrane, and cell viability, the fibrin network, and fibroblast migration were investigated. In the final analysis, statistically significant differences were found for the A-PRF group with regard to platelet-derived growth factor, transforming growth factor, epidermal growth factor and vascular endothelial growth factor at all assessment times. A difference was found with regard to bone morphogenetic protein only in the later assessment, and no differences among groups were found with regard to platelet-derived growth factor or insulin-like growth factor. The results of this systematic review and meta-analysis suggest that A-PRF has superior cellular properties and better release of growth factors compared to other platelet concentrates.

## 1. Introduction

Platelet concentrates are products obtained through the fractionation of blood plasma by centrifugation. The products of this process are cells, platelet concentrates and factors that stimulate cellular proliferation. As an easily manipulated autologous formulation, these blood products have generated considerable interest in current tissue regeneration techniques [1,2].

Different types of platelet concentrates can be obtained through changes in the centrifugation process. These formulations are classified based on the cellular products and architecture of the fibrin network. Such differences have evolved with the successive generations of platelet concentrates with the aim of resolving clinical problems through cellular properties [3]. The first generation was platelet-rich plasma, also known as leukocyte-poor platelet-rich plasma, which led to deficiencies in the healing process due to the presence of anticoagulant agents and the negative influence of the product on cellular inflammatory activation [4,5].

Beginning with the second generation, concentrates were grouped into platelet-rich fibrin (PRF), for which the presence of anticoagulants was no longer necessary. This led to leukocyte-platelet rich fibrin (L-PRF), which, along with the growth factors found in platelet concentrates, also has a high concentration of leukocytes. More recently, a new centrifugation process has been proposed, denominated low-speed centrifugation, the aim of which is to improve the distribution and quality of growth factors in the membranes used in tissue reconstruction. This novel protocol has provided advanced plasma-rich fibrin (A-PRF), and more recently an injectable formulation, which, due to its liquid structure, has different applications compared to previously produced platelet concentrates [4,6].

Advanced platelet-rich fibrin is used clinically for various purposes related to reconstructive and jaw graft surgeries. Clinical studies have shown that A-PRF achieves satisfactory results for alveolar preservation after extractions, assisting in repair and minimizing bone resorption. A-PRF has also demonstrated effectiveness as an adjuvant in the treatment of alveolar osteitis, minimizing pain, recovery repair time and associated complications. Randomized clinical trials have demonstrated the inherent advantages of A-PRF in alveolar preservation surgeries and guided bone regeneration through the osteoinductive properties and repair enhancement induced by platelet concentrates [7].

Like previous platelet concentrates, A-PRF was processed in the formulation of a membrane but with a more homogeneous distribution with regard to the quantity of platelets and leukocytes present, as well as the distribution of these cells throughout the entire membrane, except for the acellular zone, obviously. While PRF is centrifuged at 700× *g* for 12 min, A-PRF is centrifuged at 200× *g* for 14 min and I-PRF is centrifuged at 60× *g* for 3 min [7].

The satisfactory results achieved with A-PRF in tissue repair and gains are explained by biological concepts inherent to the cells found in this compound. However, much is explained by the action of growth factors, such as insulin-like growth factor 1 (IGF-1), platelet-derived growth factor (PDGF), vascular endothelial growth factor (VEGF), fibroblast growth factor (FGF), epidermal growth factor (EGF), platelet-derived epidermal growth factor (PDEGF) and proteins of the fibrin matrix, which are found in these formulations at higher concentrations than in the blood and make a direct contribution to the acceleration of the tissue regeneration process [8].

Due to the different types of platelet concentrates, questions remain with regard to the selection of which to use in accordance with the desired property in in vitro studies to provide the best assessment for the quantification of cells and growth factors. Therefore, the aim of the present systematic review and meta-analysis was to provide a detailed description of the biological and cellular properties of platelet concentrates, compiling information found in the literature to assist in the selection of concentrates based on cellular properties and the consequences of the composition.

## 2. Materials and Methods

This study was conducted following the Preferred Reporting Items for Systematic Reviews and Meta-Analyses (PRISMA, 2020) guidelines [9] and the handbook of the Cochrane Collaboration (2023) [10]. The following was the guiding question: “What are the advantages in terms of biological and cellular properties of the use of advanced platelet-rich fibrin (A-PRF)?” The PICO method was used to establish the guiding question. Population—platelet concentrates from cellular cultures; intervention—advanced platelet-rich fibrin (A-PRF); control—platelet-rich plasma (PRP), leukocyte-platelet rich fibrin (L-PRF), platelet-rich fibrin (PRF), injectable platelet-rich fibrin (I-PRF), concentrated growth factors (CGFs), pure platelet rich plasma (P-PRP), and blood clots; outcomes—cytokines, chemokines and growth factors, distribution of cells in the fibrin membrane, cell viability, fibrin network, and fibroblast migration.

### 2.1. Eligibility Criteria

The inclusion criteria were (i) in vitro studies that (ii) addressed biological or cellular aspects of A-PRF. The exclusion criterion was studies that did not compare A-PRF to another platelet concentrate of any generation or blood clot with the proposed method.

### 2.2. Information Sources and Search Criteria

Searches were performed independently by two reviewers (V.B.S.P. and D.d.S.B.) on the PubMed/Medline, Scopus, Web of Science, Embase and LILACS databases for relevant articles published to May 2023. Divergences of opinion were resolved by consulting a third reviewer (B.C.d.E.V.). Search terms combined with Boolean operators were used for each database (Table 1).

### 2.3. Article Selection

Articles were selected in two steps by two independent reviewers (V.B.S.P. and D.d.S.B.). Divergences of opinion were resolved by consulting a third reviewer (B.C.d.E.V.). Preselection was performed with the analysis of titles and abstracts based on the eligibility criteria. Preselected articles were then submitted to full-text analysis.

### 2.4. Data Extraction

Data extraction was performed by two reviewers (V.B.S.P. and D.d.S.B.) with the aid of a specific form for collecting information from the studies included in the systematic review. The following data were extracted from the articles: authors, year of publication and country of origin; cell culture; analysis; control and test groups; centrifugation protocol; and quantification of growth factors for the quantitative analysis of the data.

### 2.5. Additional Analysis

Inter-examiner agreement with regard to the articles selected was determined using the kappa statistic. Reliability between the reviewers in the article selection process was assessed assuming an acceptable limit of 80%. Disagreement at any stage was resolved by discussion and mutual decision with a third reviewer (B.C.d.E.V). The final decision was always based on the reading of the full text.

### 2.6. Risk of Bias

Risk of bias in the studies included was appraised using the QUIN method [11] for in vitro studies, with 12 quality criteria: clearly stated aims/objectives; detailed explanation of sample size calculation; detailed explanation of sampling technique; details of comparison group; detailed explanation of methodology; operator details; randomization; outcome measurement method; outcome assessor details; blinding; statistical analysis; and presentation of results. A score was attributed to each criterion and the sum total was used to classify the article as having a high, medium, or low risk of bias. Each appraisal criterion was judged as adequately specified, inadequately specified, or not specified (Appendix A).

### 2.7. Synthesis of Data

Data synthesis was performed qualitatively for seven articles [12,13,14,15,16,17,18], four of which were also included in the quantitative analysis [13,14,15,16]. Meta-analyses were performed with the aid of the Review Manager 5.4. program for the outcome “release of growth factors” in two timeframes: early (1–2 days) and later (10 to 14 days). Forests plots were created considering A-PRF (intervention) and other platelet concentrates (control). The outcome was a continuous variable expressed as mean and standard deviation, with the mean difference (MD) used as the effect measure. Inverse variance and the random model were adopted as the statistical method. A confidence level of 95% was considered and a *p*-value < 0.05 was considered indicative of statistical significance. Heterogeneity was investigated using the chi-square test, considering variability among the studies when *p* < 0.10, tau, and the I^2^ statistic. Subgroup analysis for the different platelet concentrates considered as control was performed to control heterogeneity.

### 2.8. Assessment of Certainty of the Evidence

The Grading of Recommendations, Assessment, Development and Evaluation (GRADE) approach [19] adapted for in vitro experimental studies was used to determine the certainty of the evidence of this systematic review and meta-analysis for the outcome “release of growth factors.” The assessment was based on five aspects: risk of bias, inconsistency, indirect evidence, imprecision and publication bias. The certainty of the evidence was then classified as high, moderate, low, or very low [20].

## 3. Results

### 3.1. Article Selection

After the first step of the search for titles and abstracts in the databases, a total of 589 articles were identified. Six were duplicates and were removed and 567 were excluded for not addressing the topic of interest. Fifteen articles were submitted to full-text analysis, seven of which were excluded for not meeting the eligibility criteria and one for analyzing blood of an animal origin (Appendix A). Thus, seven in vitro experimental studies investigating the biological properties of the effects of A-PRF on tissue gain, cell viability, cell counts and growth factor counts were selected for the qualitative analysis [12,13,14,15,16,17,18], four of which were also selected for meta-analysis [13,14,15,16] (Figure 1—Flowchart of article selection process). The kappa coefficient of agreement between the reviewers (V.B.S.P. and D.d.S.B.) confirmed the precision of the eligibility criteria in the selection of the articles based on the search strategy applied in each database (k = 0.92 for PubMed|Medline, Scopus, Embase, Web of Science, Cochrane Library and LILACS|bvs).

### 3.2. Characteristics of Studies

Among the seven articles selected [12,13,14,15,16,17,18], all were experimental studies with in vitro analyses comparing the biological and cellular characteristics of advanced platelet-rich fibrin (A-PRF) to at least one other platelet concentrate as control. The following were used for comparison: leukocyte- and platelet-rich fibrin (L-PRF), injectable platelet-rich fibrin (I-PRF), blood clots, platelet-rich plasma (PRP), concentrated growth factors (CGFs) and platelet-rich growth factors. Among the samples collected, seven were derived from healthy humans with no systemic comorbidities or altered blood parameters and with normal platelet counts, as displayed in the data extraction table (Appendix A). Two studies investigated fibroblast proliferation and viability in periodontal defects [17,18]. The “blood clots” variable does not fit the category of platelet concentrate but was used as a control group in some articles. All seven of the articles included had a moderate risk of bias, according to Sheth and colleagues [11]. The similarity in the appraisal results may be explained by the methodological and assessment similarities among the studies selected, as demonstrated in the risk-of-bias table (Appendix A).

### 3.3. Release of Growth Factors

Four of the studies [13,14,15,16] investigated the release of growth factors, which was the most common outcome among the studies. Growth factor counts were performed at different times, such as 1, 3, 7 and 14 days. The main growth factors investigated were insulin-like growth factor (IGF-I), platelet-derived growth factor (PDGF-BB, PDGF-AA), vascular endothelial growth factor (VEGF), bone morphogenetic protein (BMP-2) and transforming growth factor (TGF-β1). Both A-PRF and L-PRF led to a significant release of growth factors in the initial assessments, especially IGF-I. Other growth factors had a more sustained release throughout the entire assessment period, such as VEGF [13,14,15]. The best initial values were found for PRP, but A-PRF demonstrated better results after the 10th day due to its constancy. The best release results for PDGF were found for both A-PRF and I-PRF, with higher values after the third day of incubation. The release of BMP-2 was statistically similar among all platelet concentrates analyzed, with the best values found for P-PRP, intermediary values found for L-PRF, lower values found in the A-PRF group and the worst results found in the I-PRF group [16]. TGF-β, A-PRF and CGFs had similar values in the initial days, but A-PRF demonstrated superior values after seven days [13,14,15]. In the study conducted by Masuki et al. (2016) [14], the growth factor release sequence was A-PRF > CGFs > PRP > PRGF. The concentrates exhibited different release kinetics, demonstrating better results for each one, depending on the necessary release period. A-PRF had better release than traditional PRF in the long term [12].

### 3.4. Platelet Recovery

Among the studies analyzed, only one [14] investigated platelet recovery, with A-PRF demonstrating inferior results to P-PRF and L-PRF and similar results only compared to I-PRF. In the comparison to other concentrates not derived from fibrin, A-PRF (17.85) achieved better results than PRP (8.79), PRGF (2.84) and CGFs (15.51).

### 3.5. Cell Viability

Fibroblast proliferation was investigated in two studies [17,18]. Cell counts were performed 24 h after addition. A-PRF achieved better results compared to L-PRF and FGF (A-PRF = (69.2 ± 31.3%); L-PRF = (49.6 ± 24.9%; *p* = 0.00767); FGF = (26.7 ± 22.8%; *p* = 0.0078) [13]. A-PRF achieved similar fibroblast viability results compared to L-PRF and FGF, with no statistically significant differences (*p* > 0.05) [13]. In another comparison [15], A-PRF had better fibroblast proliferation values (140 ± 4%) compared to L-PRF (132.3 ± 2%), but the difference did not achieve statistical significance (*p* > 0.05) [14].

### 3.6. Cell Distribution

One study [12] investigated the distribution of cell types based on histological and histomorphometric analyses. The largest number of cells was found in the proximities of the red buffy coat (RBC) or in the buffy coat (BC). The cells present were T lymphocytes (CD3-positive cells), B lymphocytes (CD20-positive cells), stem cells (CD34-positive cells) and monocytes (CD68-positive cells). All these cells were found in the faction of red blood cells, buffy coat (BC) and distal parts. In the comparison of the L-PRF and A-PRF groups, a reduction in distribution was found from the proximal to the distal part of the clots in the control group (L-PRF). The BC was more extensive in the A-PRF group, with a more homogenous distribution and a greater presence of cells (CD15-positive neutrophilic granulocytes) in the distal part of the clots, more distant from the BC, which was not seen in the L-PRF group. The histomorphometric analysis revealed no statistically significant differences with regard to T lymphocytes (L-PRF: 12.6 ± 5%; A-PRF: 17.6 ± 9%), B lymphocytes (L-PRF: 14.6 ± 7%; A-PRF: 12.6 ± 9%) or CD34-positive stem cells (L-PRF: 17.6 ± 6%; A-PRF: 21.6 ± 11%) between the two groups in terms of the allocation of these types of cells. With the change in the centrifugation protocol for A-PRF, an increase occurred in the distribution of neutrophilic granulocytes (up to 68.6 ± 24% of the framework in the A-PRF group), whereas this type of cell was located in up to 25.6 ± 12% of the length of the clots in the L-PRF group. The statistical analysis revealed a highly significant difference in the distribution of this type of cell between the two groups (*p* < 0.05) [12].

### 3.7. Fibrin Network

Only one study [16] investigated the constitution of the fibrin network, comparing A-PRF to concentrated growth factors (CGFs) and I-PRF. The fibrin network differed in terms of thin and thick fibrillar structures among multiple regions of the concentrate. The fibrin network of CGFs and A-PRF had thicker, more organized fibers compared to the control (I-PRF). CGFs had more highly reticulated fibers compared to A-PRF. Multiple cells were found attached to the fibrin network, such as platelets and white globules, in the aggregates of A-PRF and CGFs, in contrast to the control group (I-PRF) [16].

### 3.8. Meta-Analysis

Meta-analyses were performed for the release of growth factors at two assessment times. For the release of PDGF-AA, A-PRF was superior to the control at both the 1-day assessment (MD: 322.32; 95% CI: 142.78 to 501.85; *p* = 0.0004) and 10-day assessment (MD: 211.06; 95% CI: 91.56 to 30.57; *p* = 0.0005) (Figure 2).

For PDGF-BB, no significant differences were found between the test and control groups at the 1-day assessment (MD: 55.67; 95% CI: −9.83 to 121.18; *p* = 0.10) or the assessment at 10 to 14 days (MD: −8.81; 95% CI: −26.67 to 9.05; *p* = 0.33) (Figure 3).

For the release of PDGF-AB, A-PRF was superior to the control group at the 1-day assessment (MD: 706.76; 95% CI: 481.48 to 932.04; *p* = 0.00001) and 10-day assessment (MD: 277.93; 95% CI: 101.15 to 454.72; *p* = 0.002) (Figure 4).

For the release of TGF-β, A-PRF was superior to the control group at the 1–2-day assessment (MD: 94.83; 95% CI: 72.19 to 117.47; *p* = 0.00001) as well as the 10–14-day assessment (MD: 95.90; 95% CI: 38.52 to 153.28; *p* = 0.001) (Figure 5).

A-PRF was superior to the control group for the release of VEGF at both the 1-day assessment (MD: 41.76; 95% CI: 23.44 to 60.09; *p* = 0.00001) and 10-day assessment (MD: 9,27; 95% CI: 2.35 to 16.19; *p* = 0.009) (Figure 6).

For EGF, A-PRF was superior to the control group at both the 1-day assessment (MD: 82.99; 95% CI: 17.49 to 148.49; *p* = 0.01) and 10-day assessment (MD: 28.13; 95% CI: 6.79 to 49.47; *p* = 0.010) (Figure 7).

For the release of IGF-1, the control group was superior to A-PRF at both the 1-day assessment (MD: −16.72; 95% CI: −44.43 to 10.99; *p* = 0.24) and 10-day assessment (MD: −0.98; 95% CI: −3.59 to 1.63; *p* = 0.73) (Figure 8).

No significant difference was found between the A-PRF and control group for the release of BMP-2 at the 1-day assessment (MD: −0.09; 95% CI: −0.61 to 0.42; *p* = 0.35). However, A-PRF was superior to the control group at the 14-day assessment (MD: 7.60; 95% CI: 5.53 to 9.67; *p* = 0.00001) (Figure 9).

### 3.9. Grading of Recommendations, Assessment, Development and Evaluation (GRADE)

Using the GRADE approach, the certainty of the evidence for the outcomes submitted to meta-analysis was classified as very low for BMP-2 at the 14-day assessment, PDGF-AA at the 10-day assessment, as well as TGF-β and VEGF at both assessments. The certainty of evidence was classified as low for IGF-1 and PDGF-BB at both assessments and BMP2 at the one-day assessment. Certainty of evidence was classified as moderate for EGF and PDGF-AB at both assessments and PDGF-AA at the one-day assessment (Appendix A).

## 4. Discussion

Platelet concentrates are used as additives to provide an improvement in cell migration, with the significant and lasting release of growth factors, which activate the immune system and tissue repair process, as well as improvements in the properties of the fibrin network. Second-generation concentrates are more widely used due to the better properties in comparison to platelet-rich plasma, and are therefore more often studied, as found in five of the seven articles included in the present systematic review as the control group [12,13,15,16,17].

Analyzing the evidence available in the current literature on the biological and cellular benefits of A-PRF, which is obtained through low-speed centrifugation, and considering its cell migration inductor properties, the release of growth factors, cell viability, the fibrin matrix and distribution of the cells in the framework, no previous studies have compared the properties of platelet concentrates to assist in the choice of which to use. Therefore, the present systematic review and meta-analysis was developed to provide greater scientific evidence, as most current studies focus on the clinical experience of health professionals that use such concentrates.

This study was not restricted with regard to the platelet concentrate used as control. We sought to address the properties of A-PRF in comparison to a variety of platelet and growth factor concentrates, exploring the advantages and disadvantages of each on the outcomes of interest. Therefore, this variable should not be considered a confounding factor but rather an advantage, with the use of multiple comparisons with the most varied products and their respective characteristics. To enable this assessment and diminish heterogeneity, subgroup analysis was performed for the different controls employed.

The studies described the isolation and induction of cellular formation using A-PRF in periodontal defects [17,18], periosteal cells derived from human alveolar bone [12] and through the isolation and fixation of the resulting blood clots for analysis [12,13,14]. The seven studies [12,13,14,15,16,17,18] made comparisons of the platelet concentrates separately. This is an advantage when compared to clinical studies that assess these materials, which generally involve bone substitutes that generate bias, as the properties of one can exert an influence on the analysis of the material that was used in conjunction, underscoring the importance of a comprehensive histological analysis of platelet concentrates separately [16].

The release of growth factors is one of the main advantages of platelet concentrates, as cell signaling for the tissue repair process involves elements that are abundant in PRF clots. First-generation platelet concentrates have high release rates, but the duration is short. In contrast, second-generation concentrates offer continuity in the release of growth factors for several days, leading to better long-term results [17].

Four of the studies included in the present systematic review [13,14,15,16] investigated the release of growth factors considering the main elements, such as BMP-2, which is an important factor in terms of bone tissue gain. The meta-analysis revealed no difference in the release of BMP-2 on the first day, but A-PRF achieved significantly better results on the 14th day compared to the other platelet concentrates [14,15]. PRP and CGFs achieved better results with regard to TGF-β and VEGF in the initial days, but this release diminished substantially thereafter, with L-PRF and A-PRF achieving greater release of these growth factors throughout the subsequent days [14,15]. In the quantitative analysis, A-PRF demonstrated favorable results for the release of TGF-β and VEGF at all assessment times (1, 2, 10 and 14 days).

No previous meta-analysis investigating the release of growth factors by platelet concentrates was found in the literature. Therefore, this is the first study to offer a quantitative analysis with statistical significance for this outcome. Based on the studies incorporated into the meta-analysis [13,14,15,16], A-PRF was more effective compared to the control groups at releasing all growth factors analyzed, except PDGF-BB, IGF-1 and BMP2 on the first assessment day, which demonstrates an advantage of low-speed platelet concentrates over products from previous generations and fast centrifugation in terms of this outcome.

Platelet recovery is an extremely important property, as platelets are cells that have the capacity to induce cell migration and initiate the entire tissue repair process through cell signaling. This outcome was investigated in one study [14], in which A-PRF did not achieve satisfactory results, matching the results of only I-PRF and achieving lower values compared to L-PRF and P-PRF. Second-generation platelet concentrates once again surpassed PRP in this regard (*p* < 0.05).

Fibroblast proliferation is an important aspect of tissue regeneration in the treatment of periodontal defects. A-PRF proved to be an important product in such cases, achieving the best results of all groups tested (*p* < 0.05) (L-PRF, P-PRF and blood clot) in the studies that investigated this aspect [17,18].

Cell distribution is one of the advantages of the products of low-speed centrifugation, as this change leads to a more homogeneous distribution of cells in the buffy coat, improving the properties of the clot, which is not seen in products of high-speed centrifugation (e.g., L-PRF), in which a greater concentration of cells is found at the end of the clot near the transition with the area of red blood cells [21,22,23]. One study assessed this outcome [12], and as expected, A-PRF achieved significantly better results with regard to cell distribution in comparison to L-PRF. The main cell types found were neutrophilic granulocytes (up to 68.6 ± 24%).

The fibrin network is responsible for supporting the entire cell distribution in platelet concentrates and providing a membrane property for these blood products [24,25,26]. One study [16] investigated the formation and structure of the fibrin network. A-PRF has a thicker, more reticulate network compared to CGFs and I-PRF. However, assessments of L-PRF and P-PRF are needed, as these platelet concentrates have clots similar to that of A-PRF.

The studies were classified as having a moderate risk of bias due to the lack of randomization and blinding, which ends up not being applicable to in vitro experimental studies, but leads to a reduction in the methodological quality score when the QUIN appraisal method is used [11].

In vitro experimental studies provide precise, quantified information for the analysis of the objective of our review. This design provided the necessary data for the assessment of the biological and cellular properties of A-PRF so that clinical studies with humans can be better designed with regard to the use of platelet-rich fibrin products. However, caution should be exercised in the extrapolation of the results due to the small number of studies and small samples, which generated imprecision in the results and diminished the certainty of the evidence. In the GRADE analysis [19,20], the small number of samples included in the syntheses led to a reduction in the level of certainty. This may be explained mainly by the imprecision analysis, which is directly related to sample size, as well as the broad confidence intervals.

Future studies should include larger samples to enable a more precise analysis of differences between groups and enhance the certainty of the evidence. Randomized clinical trials should be developed to assess outcomes that are directly related to the findings of the present systematic review and meta-analysis and bring findings with regard to platelet concentrates in experimental studies into clinical practice. However, the studies need to have homogeneity in the intervention and control groups and standardization in the assessment of the outcomes, which is a deficiency in clinical studies found in the current literature.

## 5. Conclusions

A-PRF achieved satisfactory results for the biological and cellular properties expected of a platelet concentrate, especially in the release of growth factors, demonstrating effective immediate and late release in comparison to other platelet concentrates (PRP, PRF and L-PRF). Low-speed centrifugation for the obtainment of A-PRF also led to improvements in terms of cell distribution and permeability.

### Protocol and Registration

This systematic review was registered in the International Prospective Register of Systematic Reviews (http://www.crd.york.ac.uk/PROSPERO) accessed on 15 April 2021, National Institute for Health Research, UK: CRD42021242984.

## Figures and Tables

**Figure 1 ijms-25-00482-f001:**
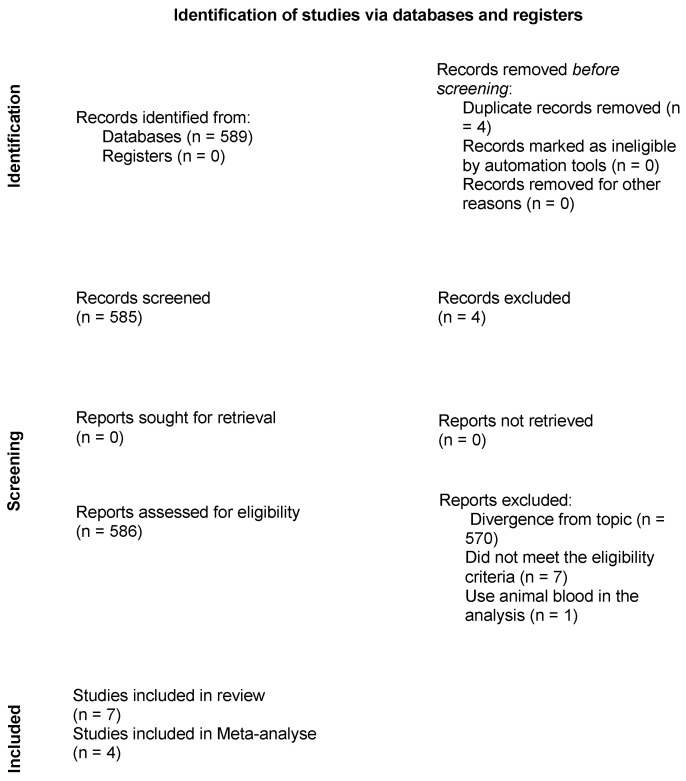
Flowchart of article selection process in accordance with the PRISMA statement.

**Figure 2 ijms-25-00482-f002:**
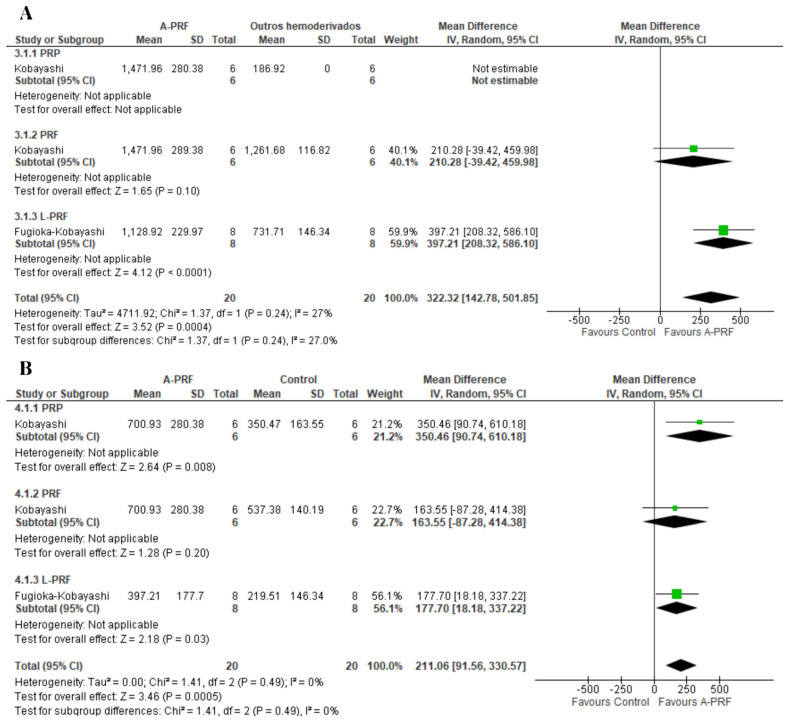
Forest plot comparing A-PRF and other platelet concentrates for release of PDGF-AA: (**A**) 1 day; (**B**) 10 days.

**Figure 3 ijms-25-00482-f003:**
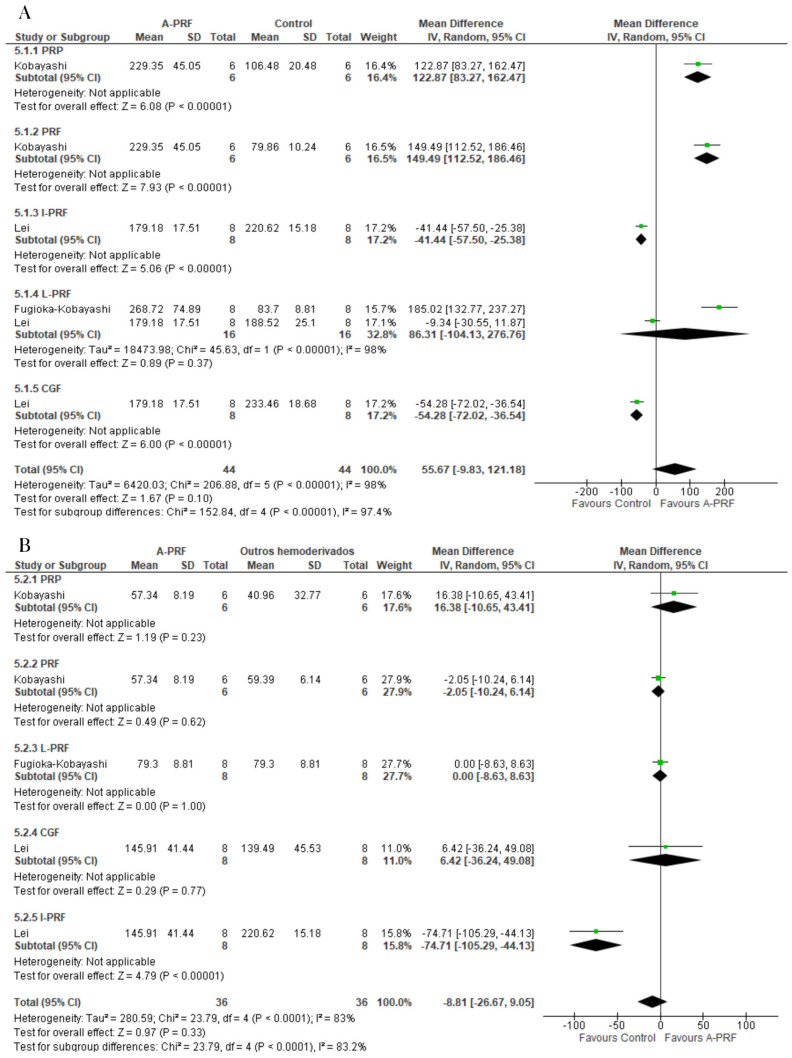
Forest plot comparing A-PRF and other platelet concentrates for release of PDGF-BB: (**A**) 1 day; (**B**) 10 to 14 days.

**Figure 4 ijms-25-00482-f004:**
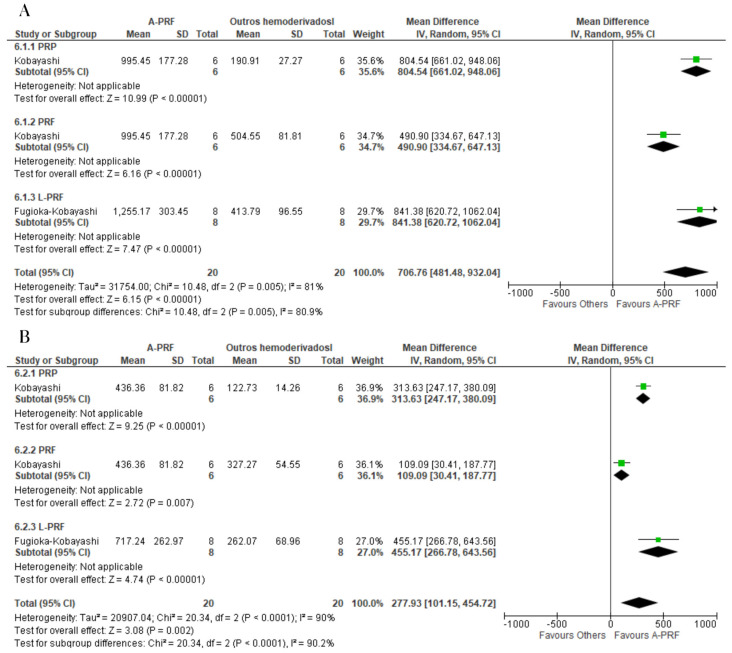
Forest plot comparing A-PRF and other platelet concentrates for release of PDGF-AB: (**A**) 1 day; (**B**) 10 days.

**Figure 5 ijms-25-00482-f005:**
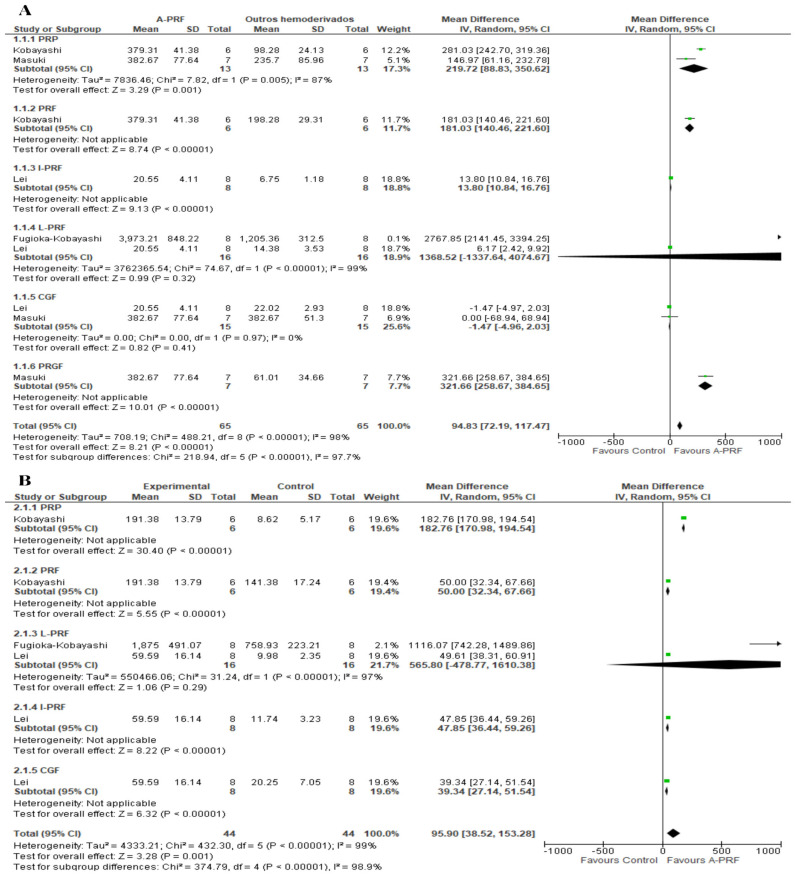
Forest plot comparing A-PRF and other platelet concentrates for release of TGF-β: (**A**) 1 to 2 days; (**B**) 10 to 14 days.

**Figure 6 ijms-25-00482-f006:**
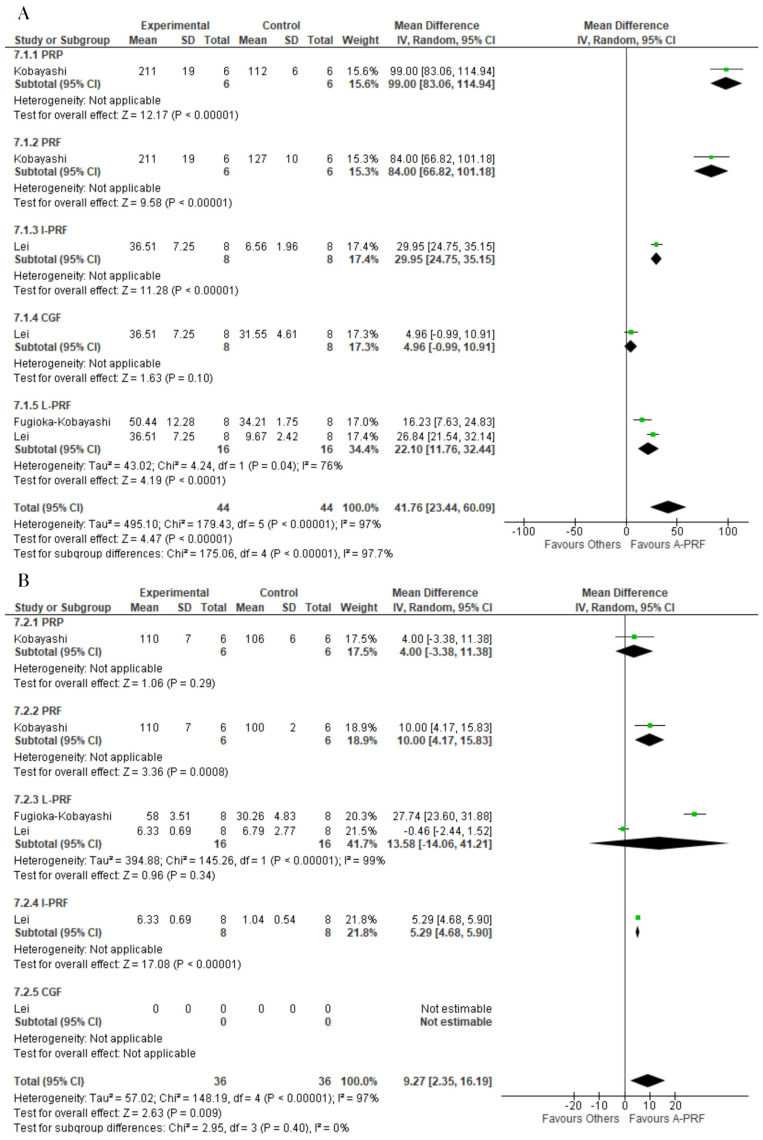
Forest plot comparing A-PRF and other platelet concentrates for release of VEGF: (**A**) 1 day; (**B**) 10 days.

**Figure 7 ijms-25-00482-f007:**
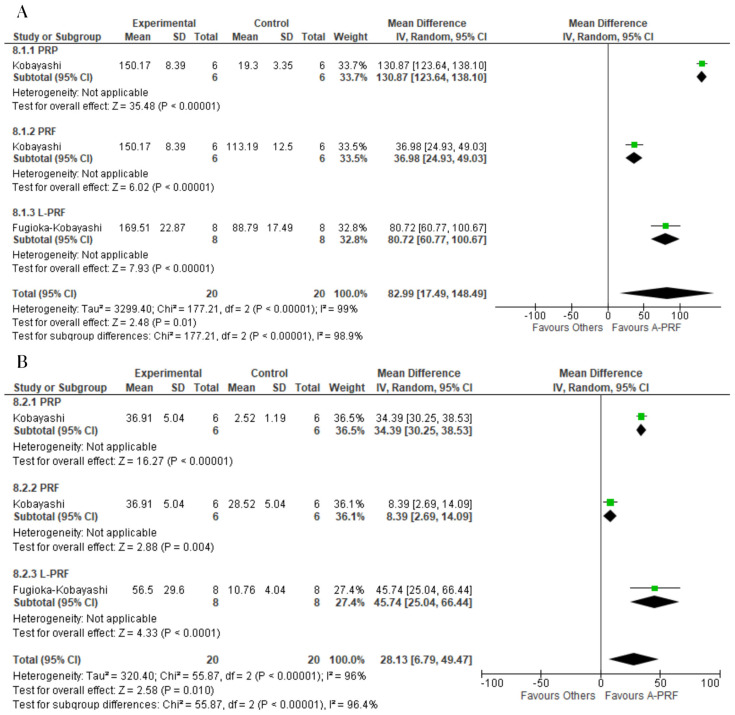
Forest plot comparing A-PRF and other platelet concentrates for release of EGF: (**A**) 1 day; (**B**) 10 days.

**Figure 8 ijms-25-00482-f008:**
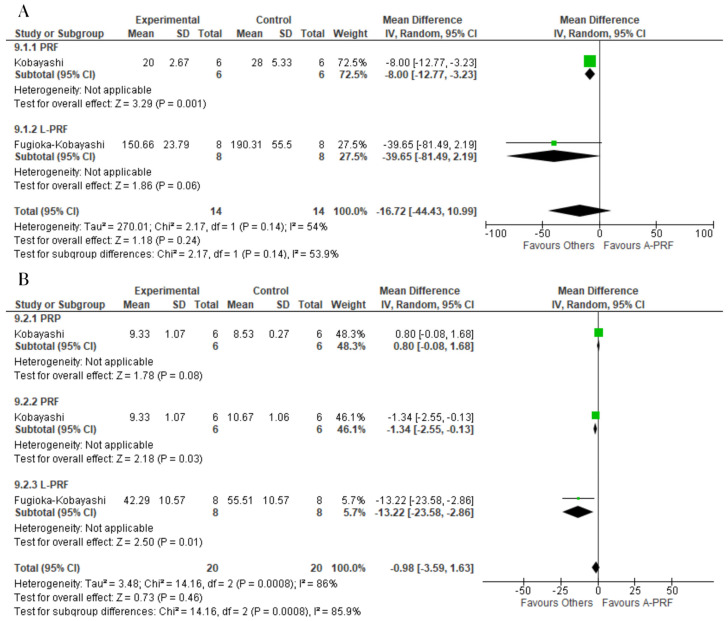
Forest plot comparing A-PRF and other platelet concentrates for release of IGF-1: (**A**) 1 day; (**B**) 10 days.

**Figure 9 ijms-25-00482-f009:**
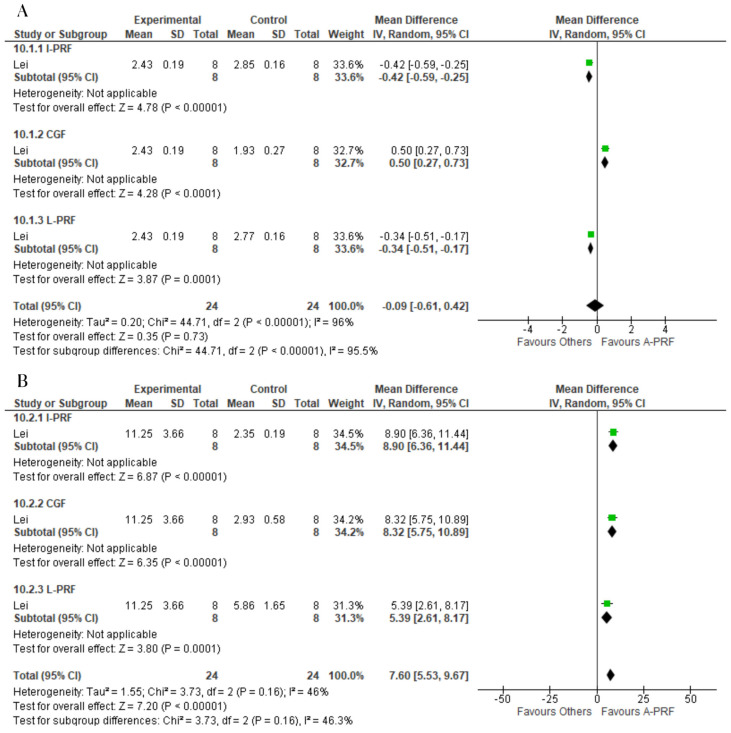
Forest plot comparing A-PRF and other platelet concentrates for release of BMP-2: (**A**) 1 day; (**B**) 10 days.

**Table 1 ijms-25-00482-t001:** Search terms combined with Boolean operators.

Terms	Database
(guided tissue regeneration OR tissue regeneration, guided OR regeneration, guided tissue OR guided regeneration, periodontal OR regeneration, periodontal guided tissue OR guided periodontal tissue regeneration OR periodontal guided tissue regeneration OR bone regeneration OR bone regenerations OR regeneration, bone OR regenerations, bone OR osteoconduction OR bone transplantation OR grafting, bone OR bone grafting OR transplantation, bone OR alveolar bone grafting OR alveolar cleft grafting OR alveolar ridge augmentation OR alveolar ridge augmentations OR augmentation, alveolar ridge OR augmentations, alveolar ridge OR ridge augmentation, alveolar OR ridge augmentations, alveolar OR mandibular ridge augmentation OR augmentation, mandibular ridge OR augmentations, mandibular ridge OR mandibular ridge augmentations OR ridge augmentation, mandibular OR ridge augmentations, mandibular OR maxillary ridge augmentation OR augmentation, maxillary ridge OR augmentations, maxillary ridge OR maxillary ridge augmentations OR ridge augmentation, maxillary OR ridge augmentations, maxillary OR sinus floor augmentation OR augmentation, sinus floor OR augmentations, sinus floor OR floor augmentation, sinus OR floor augmentations, sinus OR sinus floor augmentations OR maxillary sinus floor augmentation OR sinus augmentation therapy OR augmentation therapies, sinus OR augmentation therapy, sinus OR sinus augmentation therapies OR therapies, sinus augmentation OR therapy, sinus augmentation OR regeneration OR guided tissue regeneration, periodontal) AND (A-PRF OR advanced platelet-rich fibrin) AND (L-PRF OR platelet-rich fibrin OR fibrin, platelet-rich OR platelet rich fibrin OR L-PRF OR leukocyte- and platelet-rich fibrin OR leucocyte and platelet rich fibrin)	PubMed|Medline, Web of Science
(ALL (tissue AND regeneration) OR ALL (bone AND conduction) OR ALL (bone AND graft) OR ALL (alveolar AND ridge AND augmentation) OR ALL (sinus AND floor AND augmentation) AND ALL (platelet-rich AND fibrin) OR ALL (platelet-rich AND fibrin AND matrix)) AND (LIMIT-TO (DOCTYPE, “ar”))	Scopus
((“tissue regeneration” OR “bone conduction” OR “bone graft” OR “alveolar ridge augmentation” OR “sinus floor augmentation”) AND (“platelet-rich fibrin” OR “platelet rich fibrin matrix”)) AND “article”/it	Embase
(tissue regeneration OR bone conduction OR bone graft OR alveolar ridge augmentation OR sinus floor augmentation) AND (platelet-rich fibrin OR platelet rich plasma)	LILACS|bvs

## Data Availability

Not applicable.

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
