# Peer review of "Biological and Cellular Properties of Advanced Platelet-Rich Fibrin (A-PRF) Compared to Other Platelet Concentrates: Systematic Review and Meta-Analysis"

_ijms, 2023, doi:10.3390/ijms25010482_

Round 1

Reviewer 1 Report

Comments and Suggestions for Authors

This review (actually a systematic literature review +  meta-analysis of 4 studies)  compares the biological and cellular properties of advanced platelet-rich fibrin (A-PRF) and other platelet concentrates. These are used in experimental and clinical healing and tissue regeneration studies. The results of this systematic review and meta-analysis suggest that A-PRF has superior cellular properties, for example increased release of growth factors compared to other platelet concentrates. (Increased release of PDGF-AB, PDGF-AA, TGF-β, EGF, VEGF, and BMP at later times.)

This is  a well presented and well-written study which is biomedically and clinically (wound healing and others)  of broad interest. I have only minor questions/and suggestions.

1) Are advanced platelet-rich-fibrin (A-PRF) already used clinically, and what for? Should be mentioned in the paper.

2) Are such fibrin/platelet concentrates associated with any infection risks?

3) What do the authors consider as very important clinical studies, which are also  doable?

4) Have double blinded  studies performed , even with a true placebo (no platelets) ?  

Comments on the Quality of English Language

English is fine for this referee.  

Author Response

From: Vinicius Balan Santos Pereira and co-authors

Date: December 25, 2023

Subject: International Journal of Molecular Sciences: Response letter to Reviewer 1

Manuscript ID: ijms-2752785

Type of manuscript: Review

Title: Biological and Cellular Properties of Advanced Platelet-rich Fibrin (A-PRF) Compared to Other Platelet Concentrates: Systematic Review and Meta-Analysis

To: Prof. Dr. Maurizio Battino (Editors-in-Chief) | Prof. Dr. Andreas Taubert (Section Editor-in-Chief – Materials Science Section)

Point-by-point response to the comments

REVIEWER REPORTS

Reviewer Comments:

Reviewer 1

This review (actually a systematic literature review +  meta-analysis of 4 studies)  compares the biological and cellular properties of advanced platelet-rich fibrin (A-PRF) and other platelet concentrates. These are used in experimental and clinical healing and tissue regenerationstudies. The results of this systematic review and meta-analysis suggest that A-PRF has superior cellular properties, for example increased release of growth factors compared to other platelet concentrates. (Increased release of PDGF-AB, PDGF-AA, TGF-β, EGF, VEGF, and BMP at later times).

This is  a well presented and well-written study which is biomedically and clinically (wound healing and others)  of broad interest. I have only minor questions/and suggestions.

Response: Thank you for giving us the opportunity to submit a revised draft of our manuscript titled " Biological and Cellular Properties of Advanced Platelet-rich Fibrin (A-PRF) Compared to Other Platelet Concentrates: Systematic Review and Meta-Analysis" to the International Journal of Molecular Sciences. We appreciate the time and effort that you and the reviewers have dedicated to providing your valuable feedback on our manuscript. We are grateful to the reviewers for their insightful comments on our paper. All requests were accepted. We have highlighted the changes within the manuscript.

Here is a point-by-point response to the reviewers’ comments and concerns.

Comments from Reviewer 1

Comment 1:

1) Are advanced platelet-rich-fibrin (A-PRF) already used clinically, and what for? Should be mentioned in the paper.

Response: We appreciate the reviewer's comment. As requested, a randomized controlled clinical trial (recently published) supporting this point have been integrated to the Introduction (highlighted in red color).

Comment 2:

2) Are such fibrin/platelet concentrates associated with any infection risks?

Response: Platelet concentrates, such as A-PRF, are products derived from the centrifugation of autogenous blood for individual use, eliminating the risk of contamination by infectious diseases. Blood collection is similar to routine hematological tests. Furthermore, the clinical management of these blood products is carried out using sterile surgical instruments and sterile disposable materials. Therefore, the risk of infection is not expected. It is important to emphasize that evidence-based practice carried out by qualified professionals, under favorable conditions, does not represent any infection risks.

Reference: Ghanaati S, Booms P, Orlowska A, Kubesch A, Lorenz J, Rutkowski J, Landes C, Sader R, Kirkpatrick C, Choukroun J. Advanced platelet-rich fibrin: a new concept for cell-based tissue engineering by means of inflammatory cells. J Oral Implantol. 2014 Dec;40(6):679-89. doi: 10.1563/aaid-joi-D-14-00138.

Comment 3 and 4:

3) What do the authors consider as very important clinical studies, which are also doable?

4) Have double blinded  studies performed, even with a true placebo (no platelets) ?

Response: Randomized controlled clinical trials, preferably triple-blinded and simple-arm (paired), represent the primary studies with the highest level of clinical evidence. However, it may be difficult to ensure blinding of the surgeon (intervention) and the patient in this type of study. This systematic review addresses preclinical evidence on a clinically relevant topic, where the biological basis remains uncertain to date. In this case, only the synthesis and analysis of preclinical data is capable of guiding new laboratory and clinical research and supporting critical analysis or understanding of clinical results. In this context, it is also possible to question clinical tests with platelet concentrates without adequate preclinical evidence. Furthermore, the scarce literature and the methodological dissimilarity of clinical studies on the topic make it difficult to synthesize evidence to guide clinical decision-making, in which it is necessary to establish the level of evidence and the degree of recommendation (GRADE).

Comment 5:

Comments on the Quality of English Language.

English is fine for this referee.

Response: We appreciate the reviewer's comment.

The authors remain at the disposal of the International Journal of Molecular Sciences for any additional review or correction of the manuscript.

We look forward to responding to any further questions and comments you may have.

Thank you for considering our manuscript for publication.

Sincerely,

Vinicius Balan Santos Pereira and co-authors

Reviewer 2 Report

Comments and Suggestions for Authors

In the systematic review and meta-analysis entitled: “Biological and Cellular Properties of Advanced Platelet-rich Fibrin (A-PRF) Compared to Other Platelet Concentrates: Systematic Review and Meta-Analysis”, by Dr Pereira et al., the Authors searched biomedical literature databases, retrieved and analysed papers reporting data on biological activity of various preparations of platelet origin obtained using different laboratory methods. I think that idea of study is interesting and the results are potentially valuable, however I have several both major as well as minor comments. In general, the quality of presentation should be significantly improved.

Major comments

1. The grey literature, which is not by definition scientific papers, should not be included in systematic review/meta-analysis.

2. Lines 54-56: RPM (revolutions per minute) is not an objective measure of centrifugation speed. Depending on the rotor diameters, the same RPM values could give various relative centrifugal force (RCF, expressed in g). The RCF values, as the objective and comparable parameter, should be given here.

Minor comments:

1. Abstract, line 15: databased -> databases

2. Some abbreviations in the abstract are unclear and since they are not wiritten in full, the abstract is difficult to be understood, e.g. PDGF-AB, PDGF-AA, BMP

3. Line 41 a word „fibrin” is missing before (PRF)

4. The quality (resolution) of the figures is poor. It makes difficult to analyse the results correctly.

Comments on the Quality of English Language

The paper is interesting however a significant improvement of the quality of presentation should be made before the publication. Please see my detailed remarks available to the Authors.

Author Response

From: Vinicius Balan Santos Pereira and co-authors

Date: December 25, 2023

Subject: International Journal of Molecular Sciences: Response letter to Reviewer 2

Manuscript ID: ijms-2752785

Type of manuscript: Review

Title: Biological and Cellular Properties of Advanced Platelet-rich Fibrin (A-PRF) Compared to Other Platelet Concentrates: Systematic Review and Meta-Analysis

To: Prof. Dr. Maurizio Battino (Editors-in-Chief) | Prof. Dr. Andreas Taubert (Section Editor-in-Chief – Materials Science Section)

Point-by-point response to the comments

REVIEWER REPORTS

Reviewer Comments:

Reviewer 2

In the systematic review and meta-analysis entitled: “Biological and Cellular Properties of Advanced Platelet-rich Fibrin (A-PRF) Compared to Other Platelet Concentrates: Systematic Review and Meta-Analysis”, by Dr Pereira et al., the Authors searched biomedical literature databases, retrieved and analysed papers reporting data on biological activity of various preparations of platelet origin obtained using different laboratory methods. I think that idea of study is interesting and the results are potentially valuable, however I have several both major as well as minor comments. In general, the quality of presentation should be significantly improved.

Response: Thank you for giving us the opportunity to submit a revised draft of our manuscript titled " Biological and Cellular Properties of Advanced Platelet-rich Fibrin (A-PRF) Compared to Other Platelet Concentrates: Systematic Review and Meta-Analysis" to the International Journal of Molecular Sciences. We appreciate the time and effort that you and the reviewers have dedicated to providing your valuable feedback on our manuscript. We are grateful to the reviewers for their insightful comments on our paper. All requests were accepted. We have highlighted the changes within the manuscript.

Here is a point-by-point response to the reviewers’ comments and concerns.

Comments from Reviewer 1

Major comments

Comment 1:

  1. The grey literature, which is not by definition scientific papers, should not be included in systematic review/meta-analysis.

Response: The authors agree with the reviewer's comment. Although grey literature was consulted in the initial data search, no data were included in the qualitative synthesis or meta-analysis of the review. As requested, the authors removed grey literature content from the manuscript (Materials and methods and Results sections).

Comment 2:

  1. Lines 54-56: RPM (revolutions per minute) is not an objective measure of centrifugation speed. Depending on the rotor diameters, the same RPM values could give various relative centrifugal force (RCF, expressed in g). The RCF values, as the objective and comparable parameter, should be given here.

Response: We agree and appreciate the reviewer's suggestion. RCF is now expressed in G-force (G) in the Results and supplementary tables (highlighted in yellow color).

Minor comments

Comment 1:

  1. Abstract, line 15: databased -> databases

Response: As requested, the term "databased" has been replaced by "databases" (highlighted in green color).

Comment 2:

  1. Some abbreviations in the abstract are unclear and since they are not written in full, the abstract is difficult to be understood, e.g. PDGF-AB, PDGF-AA, BMP

Response: As requested, the abbreviations were replaced by the full name (highlighted in pink color).

Comment 3:

  1. Line 41 a word “fibrin” is missing before (PRF).

Response: As requested, the word “fibrin” was integrated to the Introduction, before PRF) (highlighted in grey color).

Comment 4:

  1. The quality (resolution) of the figures is poor. It makes difficult to analyse the results correctly.

Response: As requested, the quality (resolution) of the figures has been improved.

Comments on the Quality of English Language:

The paper is interesting however a significant improvement of the quality of presentation should be made before the publication. Please see my detailed remarks available to the Authors.

Response: We apologize for that. The authors contacted a native English speaker to proofread the manuscript.

The authors remain at the disposal of the International Journal of Molecular Sciences for any additional review or correction of the manuscript.

We look forward to responding to any further questions and comments you may have.

Thank you for considering our manuscript for publication.

Sincerely,

Vinicius Balan Santos Pereira and co-authors

Round 2

Reviewer 2 Report

Comments and Suggestions for Authors

The Authors have responded for all the my comments and the paper has been significantly improved. I have no further comments.